# Wireless Passive Microwave Antenna-Integrated Temperature Sensor Based on CSRR

**DOI:** 10.3390/mi13040621

**Published:** 2022-04-15

**Authors:** Hairong Kou, Libo Yang, Xiaoyong Zhang, Zhenzhen Shang, Junbing Shi, Xiaoli Wang

**Affiliations:** 1Department of Intelligence and Automation, Taiyuan University, Taiyuan 030032, China; zhangxiaoyong@tyu.edu.cn (X.Z.); shangzhenzhen@tyu.edu.cn (Z.S.); shijunbing@tyu.edu.cn (J.S.); 2Jinxi Industries Group Co., Ltd., Taiyuan 030024, China; b1606009@st.nuc.edu.cn

**Keywords:** temperature sensor, CSRR, wireless and passive, harsh environment, multi-site

## Abstract

A novel, wireless, passive substrate-integrated waveguide (SIW) temperature sensor based on a complementary split-ring resonator (CSRR) is presented for ultra-high-temperature applications. The temperature sensor model was established by using the software of HFSS (ANSYS, Canonsburg, PA, USA) to optimize the performance. This sensor can monitor temperature wirelessly using the microwave backscatter principle, which uses a robust high-temperature co-fired ceramic (HTCC) as the substrate for harsh environments. The results are experimentally verified by measuring the S (1,1) parameter of the interrogator antenna without contact. The resonant frequency of the sensor decreases with the increasing temperature using the dielectric perturbation method, which changes from 2.5808 to 2.35941 GHz as the temperature increases from 25 to 1200 °C. The sensitivity of the sensor is 126.74 kHz/°C in the range of 25–400 °C and 217.33 kHz/°C in the range of 400–1200 °C. The sensor described in this study has the advantages of simple structure, higher quality and sensitivity, and lower environmental interference, and has the potential for utilization in multi-site temperature testing or multi-parameter testing (temperature, pressure, gas) in high-temperature environments.

## 1. Introduction

Temperature measurement [1,2,3] has great significance in the fields of aerospace [4,5], industrial production [6,7], etc. For example, the high heat flow generated by aeroengines during high-speed operation means the surface temperature of the turbine blade exceeds 1000 °C [8], which causes the turbine blade to be easily damaged. The repair of turbine blades is very expensive and complicated, so accurate monitoring of the surface temperature of the turbine blades is especially important for the safe operation of the engine. HTCC materials can withstand high temperature and corrosion, and have become one of the most promising ceramic materials in high-temperature fields such as aerospace and industrial production. Traditional silicon-based sensors [9,10] have received extensive attention due to their small size and high integration, but their development in the aviation field has been constrained by their high-temperature performance and wired test method.

Wireless and passive sensors can solve the problem of lead failure with wired sensors in high-temperature environments, and have unparalleled advantages in harsh engine environments. The wireless and passive methods mainly consist of three types: inductance–capacitance coupling (LC) [11,12,13], surface acoustic wave technology (SAW) [14,15,16], and microwave backscattering technology [17,18,19]. Tan et al. [11] designed a LC temperature sensor, fabricated on multilayer high-temperature cofired ceramic (HTCC) tapes, which can work in the temperature range of 20–900 °C with a sensitivity of 5.22 kHz/°C. It is a pity that the LC sensors are susceptible to disturbances from the metal environment. Shu et al. [14] developed a langasite SAW temperature sensor, which was tested up to 600 °C. The SAW sensors require an external antenna to transmit the signal, which will cause failure of the connection between the sensor and antenna in high-temperature environments. Cheng et al. [19] reported a novel wireless passive temperature sensor based on microwave backscattering with a reflective patch, which can work up to 1050 °C with a high sensitivity of 580 kHz/°C. Microwave backscattering technology has the advantages of high-quality factor and sensitivity, strong anti-interference ability, and can work in metal environments, which is a promising new telemetry for the applications of engine propulsion control and health monitoring systems.

In this paper, we propose a novel, wireless, passive substrate-integrated waveguide (SIW) temperature sensor based on a complementary split-ring resonator (CSRR). The CSRR structure can receive and emit the echo signal to an interrogator antenna without contact based on the microwave backscatter principle. The sensor was simulated to work at 2.5 GHz with high performance by the HFSS, and was fabricated using a robust high-temperature co-fired ceramic (HTCC) as the substrate for high-temperature applications. The dielectric constant increases with the increasing temperature, which causes a decrease of the resonant frequency of the sensor using a dielectric perturbation method. By analyzing the return loss (S11) of the interrogator antenna reflected by CSRR, the resonant frequency of the sensor will be obtained, and the relationship between the environment and the sensor can also be obtained.

## 2. Sensor Design and Fabrication

### 2.1. Sensing Theory of Sensor

Antenna-integrated sensing technology plays an important role in the wireless and passive system. The temperature sensor consists of two parts: a microwave cavity resonator and a complementary resonant ring, as shown in Figure 1a. The microwave cavity resonator is formed by SIW [20,21] and coating an alumina substrate with platinum (Pt). The SIW is composed of the upper and lower metal surfaces and metallized hole-array, which has low radiation loss and insertion loss. The SIW structure can condense electromagnetic waves inside the resonant cavity. Owing to the gaps between the adjacent discrete metal hole-array, the longitudinal propagative surface current will be cut off, meaning the transverse magnetic (TM) model cannot propagate in the SIW. The transverse electric (TE) mode propagates in the SIW, forming a current flowing along the metal holes, as shown in Figure 1b. According to [22], when the parameters of metallized holes satisfy the following conditions, electromagnetic waves leaking from the metallized holes are negligible:(1)$$p<0.2\lambda_{g},\;p<4D,D<0.2W_{eff}$$
(2)$$W_{eff}=W-1.08\frac{D^{2}}{p}+0.1\frac{D^{2}}{W}$$
where *λ_g_* is the guided wavelength, *W_eff_* represents the equivalent length and width of the substrate-integrated waveguide, *W* and *L* represent the length and width of the substrate, respectively, *D* is the diameter of the metal holes, and *p* is the center-to-center space between the two adjacent metal holes.

The sensing mechanism shown in Figure 2 is suitable for harsh environments of high temperature and high spin because of the wireless passive sensing technique, which consists of a waveguide antenna and a temperature sensor. The waveguide antenna WR-340 (width of cross-section: 86.36 mm, height of cross-section: 43.18 mm) acts as an interrogator antenna to emit electromagnetic signals from 2.2 to 3.3 GHz during the simulation process by the high-frequency simulation structure (HFSS). The CSRR structure is embedded on the surface of the SIW resonator, which is used as a response antenna to couple electromagnetic signals into the SIW resonator. When the transmitted signal is the same as the resonant frequency of the temperature sensor, the CSRR will reflect self-resonance frequency components back to the waveguide antenna, as shown in Figure 2b. When the ambient temperature increases, the dielectric constant of the alumina ceramic increases accordingly, which causes a decrease in the capacitance between the upper and lower metal surfaces, resulting in a decrease of resonant frequency of the temperature sensor, according to Equations (3) and (4). By analyzing the return loss (S11) of the waveguide antenna reflected by CSRR, the resonant frequency of the sensor will be obtained, and the relationship between the environment and the sensor can be deduced by the return loss.
(3)$$f=\frac{1}{2\pi \sqrt{LC}}$$
(4)$$C=\frac{\varepsilon_{0}\varepsilon_{r}A}{d}$$
where *ε*_0_ is the permittivity of the vacuum, *ε_r_* is the relative permittivity of the dielectric layer HTCC, *A* is the plate area, and *d* is the plate spacing.

### 2.2. Design and Simulation of the Sensor

In order to improve the signal coupling between the waveguide antenna and the temperature sensor, the model illustrated in Figure 2a was established by HFSS software to simulate and optimize the parameters of the sensor. Hence, we chose the width, a, of CSRR, the gap, b, between outer and inner resonant rings, the length, t, of CSRR, and the distance, h, between the waveguide antenna and the sensor to improve the impedance matching, as shown in Figure 3a–d. We attached importance to the impedance matching between the waveguide antenna and the sensor because good impedance matching enables a higher sensitivity of the test and remote wireless transmission. Since impedance matching is associated with the return loss, we can obtain the optimized parameters of the sensor at the resonant frequency of 2.5 GHz by comparing the return loss of the corresponding simulated curve. At present, 2.5 GHz is a world interoperability for microwave access (WiMAX) bands, and has thus obtained wide application. It can be seen from Figure 3d that the optimal coupling distance between the sensor and the antenna is 20 mm, and when the coupling distance is greater than 20 mm, the signal of the sensor is weakened.

The final parameters of the sensor that satisfied the WiMAX band are listed in Table 1.

Figure 4 displays the magnitude of the electric field distribution and magnetic field distribution inside the CSRR-loaded SIW resonant cavity. The intense electromagnetic field distribution is mainly concentrated on the CSRR structure. The measurement of pressure will be achieved by setting pressure-sensitive elements within the substrate and located inside the CSRR structure using electromagnetic field perturbation.

To study the strongest radiation direction and radiation efficiency, the 3D gain sphere and radiation pattern were simulated by HFSS, as shown in Figure 5. Figure 5a shows that the strongest radiation direction is directly above the sensor. The longest transmission distance can be achieved by the interrogator antenna facing toward the sensor. Theta (θ) is the angle with the positive direction of the z-axis. Figure 5b shows the normalized gain with different θ, from −180° to 180°, under Phi = 0 degree When θ = 0, that is the positive direction of the z-axis, the maximum gain is 8.45 dB.

To vividly understand the relationship between the temperature and the sensor, the resonant frequency of the sensor was simulated with the different dielectric constants, as shown in Figure 6a.

When temperature increased from 25 to 1200 °C, the dielectric constant of alumina ceramics increased from 9.8 to 12, according to [23]. We set the dielectric constant as a variable, which increased from 9.8 to 11.9 at a step of 0.3. The resonant frequency of the sensor decreased from 2.4949 to 2.2289 GHz, which is up to 266 MHz. The curve of resonant frequency versus the dielectric constant is plotted in Figure 6b, which presents an approximative linear change and indicates that the sensor is suitable for ultra-high-temperature measurement.

### 2.3. Multi-Site and Netted Implementation

Figure 7 illustrates the application scenario of the multi-site temperature measurement in aeroengine blades. In this work, a CSRR was used as the core structure of the sensor. Sensors of different resonant frequencies can be designed by adjusting the length of a CSRR (L1, L2, and L3 are the lengths of the outer ring of each CSRR, respectively). The longer the outer ring side length, the lower the resonant frequency. These can be used for multi-site, networked testing. The three sensors loaded with different CSRR structures were employed for multiple discrete frequency bands to realize the multi-site and netted temperature testing. The interrogation antenna, such as the horn antenna, connected to a network analyzer, illuminates the sensors 1, 2, and 3 with a signal containing the corresponding self-resonance frequency. When the external environment temperature changes, the resonance frequency of each sensor can be changed in a certain range. This method can simultaneously realize multi-site temperature measurements.

### 2.4. Sensor Fabrication

The sensor was fabricated using the 99% alumina ceramic of 28 × 28 mm as a substrate. Then, the cylindrical array vias on substrate were realized by laser drilling technology. The CSRR structure was printed on top of the substrate with platinum paste using screen-printing technology. Subsequently, the printed sensor was placed in the high-temperature furnace at 100 °C for 20 min for drying. The electrode of the bottom substrate was printed and dried as in the previous steps. The next step was filling cylindrical array vias using the platinum paste to realize the connection of upper and lower metal surfaces. Finally, the fabricated sensor was placed into the sintering furnace to metalize the platinum paste, as was demonstrated in our previous work [24,25], as shown in Figure 8b. The fabricated sensor is shown in Figure 8a.

## 3. Measurement and Discussion

To characterize the performance of the fabricated sensor, we built a wireless temperature test platform, as shown in Figure 9, which consists of a maffle furnace, a network analyzer (N5061B, Santa Clara, CA, USA), a temperature sensor, a coplanar waveguide antenna, and LabView software. The maffle furnace provides a high-temperature environment for real-time testing. The temperature sensor was placed in the maffle furnace, and the coplanar waveguide antenna connected a network analyzer with a coaxial line was faced toward the sensor, which acts as an integrated antenna to emit the electromagnetic wave signal to the sensor and receive the reflected electromagnetic signals transmitted by the temperature sensor. The LabView software was connected to the network analyzer for real-time temperature data collection. When the environmental temperature changes, the resonant frequency of the sensor changes accordingly, which can be obtained from the LabView software.

Figure 10 shows the measured results of the sensor under a temperature range of 25–1200 °C. The peak of the measured curves corresponded to the resonant frequency point of the sensor. As shown in Figure 10a, when the temperature increased, the resonant frequencies of the temperature sensor drifted towards the lower frequency. That is because the increasing temperature causes the dielectric constant to increase, resulting in the resonant frequency decreasing. The resonant frequency of the sensor decreased from 2.5808 to 2.35941 GHz as the temperature increased from 25 to 1200 °C, which is up to 221.39 MHz. Figure 10b shows the linear fitting of resonant frequency under a temperature range of 25–1200 °C. In order to accurately analyze the sensitivity of the temperature sensor, we divided the curve into two ranges: low-temperature range (25–400 °C) and high-temperature range (400–1200 °C). It can be seen from Figure 10b that the sensor had a higher sensitivity in the high-temperature range than the low-temperature range. The sensitivity of the sensor was 126.74 kHz/°C in the range of 25–400 °C and 217.33 kHz/°C in the range of 400–1200 °C. In Figure 10c,d, the CSRR-based sensor had a maximum of 0.033 GHz nonlinearity errors in the range of 25–400 °C, corresponding to a 0.102% nonlinearity error, and a maximum of 0.0108 GHz nonlinearity errors in the range of 400–1200 °C, corresponding to a 0.46% nonlinearity error.

Figure 10e illustrates the curves of measured and simulated relative resonant frequency versus temperature under the temperature range of 25–1200 °C. The variation of simulated resonant frequency was 266 MHz in the temperature range of 25–1200 °C, and the measured resonant range was 221.4 MHz. There is a certain deviation between the measured sensitivity and the simulated sensitivity, mainly due to the imprecise value of the dielectric constant of alumina ceramics with different temperatures, and thus it is necessary to research the thermal performance of alumina ceramics in future work. In order to verify the repeatability of the temperature sensor, we have carried out three tests, as shown in Figure 10f. The test results showed that the sensor has good repeatability and can be used for testing temperature parameters in the ultra-high-temperature environment. It has broad application prospects in aerospace, energy exploitation, and industrial fields.

## 4. Conclusions

This paper proposed aa wireless and passive microwave temperature sensor based on a complementary split-ring resonator. A wireless, passive, CSRR structure-integrated SIW resonator working in the 2.5 GHz frequency band was achieved by optimizing the sensor parameters using HFSS simulation software. The sensor was fabricated using the laser-etching process and screen-printing technology, which was experimentally verified to monitor the temperature without contact in a harsh environment. Furthermore, the temperature can work up to 1200 °C with the sensitivity of 126.74 kHz/°C in the range of 25–400 °C and 217.33 kHz/°C in the range of 400–1200 °C. In the future, this microwave sensor can also realize the wireless acquisition of the pressure signal in high-temperature environments by adding a pressure-sensitive element in the substrate, which is promising for multi-site temperature testing or multi-parameter testing (temperature, pressure, gas, etc.) in high-temperature environments.

## Figures and Tables

**Figure 1 micromachines-13-00621-f001:**
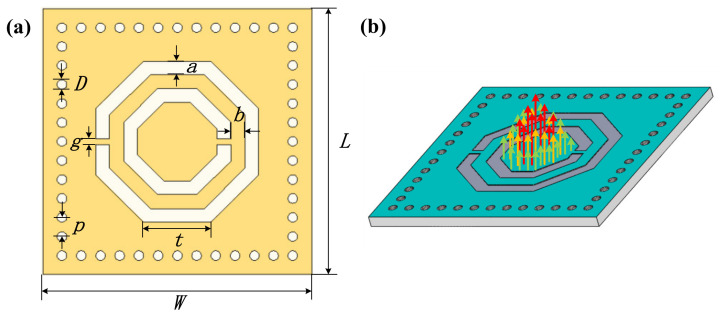
Model of the proposed CSRR integrated sensor. (**a**) The physical and geometric parameters of the CSRR integrated temperature sensor. (**b**) Electric field distribution.

**Figure 2 micromachines-13-00621-f002:**
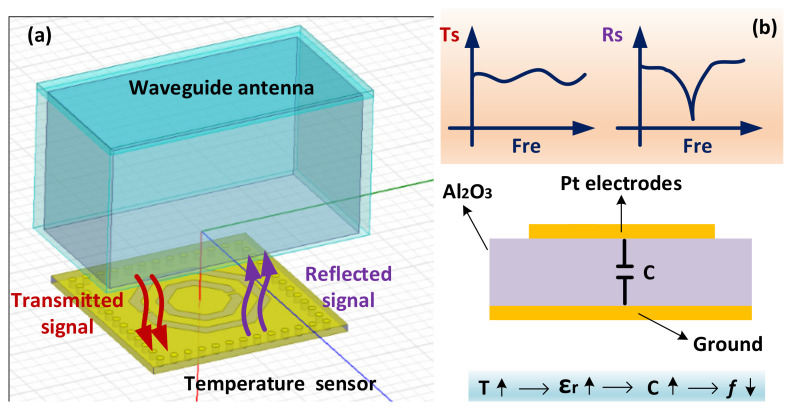
Wireless sensing mechanism of the sensor. (**a**) Transmission principle. (**b**) Working principle of the sensor.

**Figure 3 micromachines-13-00621-f003:**
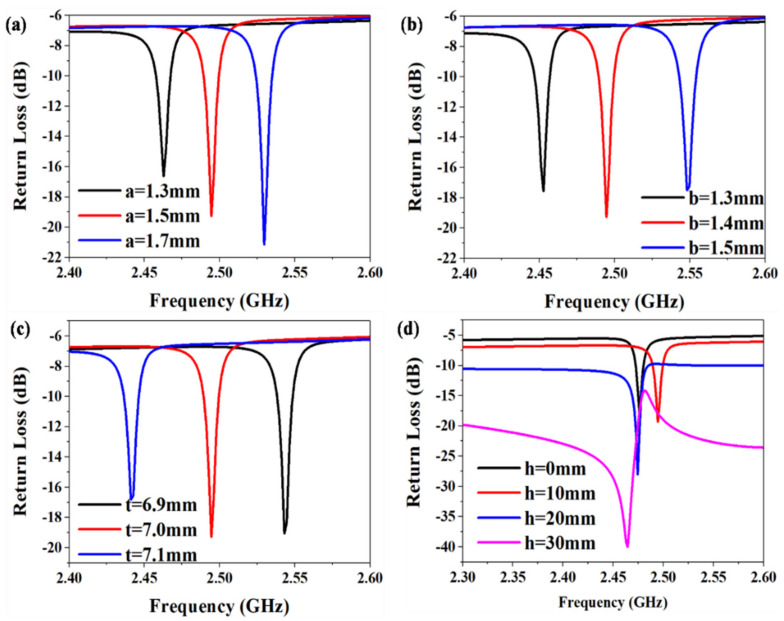
HFSS simulated results of: (**a**) width, a, of the slot antenna, (**b**) gap, b, of CSRR, (**c**) length, t, of the slot antenna, and (**d**) distance, h, between the waveguide antenna and the sensor.

**Figure 4 micromachines-13-00621-f004:**
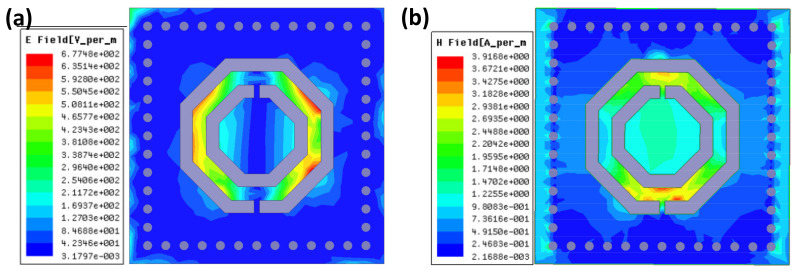
(**a**) Electric field distribution. (**b**) Magnetic field distribution.

**Figure 5 micromachines-13-00621-f005:**
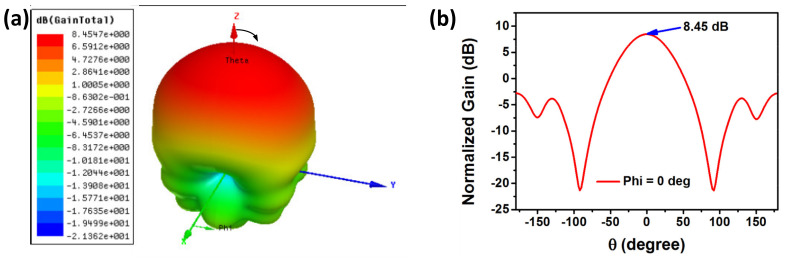
(**a**) 3D gain sphere. (**b**) Simulated radiation pattern when Phi = 0 degree.

**Figure 6 micromachines-13-00621-f006:**
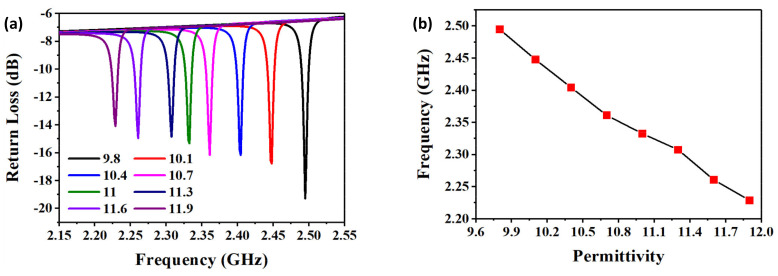
(**a**) Resonant frequency versus return loss with different dielectric constants. (**b**) Frequency fitting curve with different dielectric constants.

**Figure 7 micromachines-13-00621-f007:**
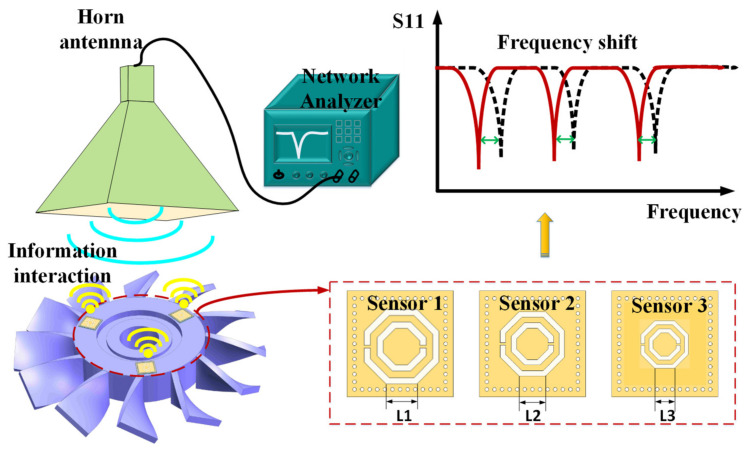
The schematic diagram of the multi-site and netted implementation.

**Figure 8 micromachines-13-00621-f008:**
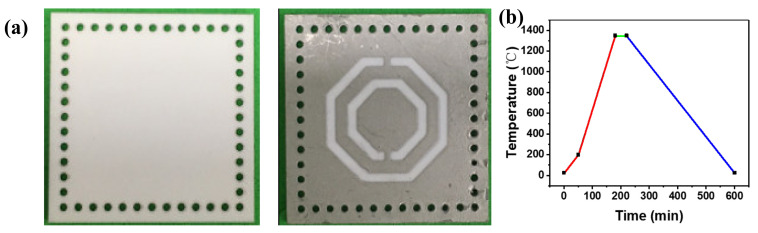
(**a**) Alumina ceramic substrate and fabricated sensor. (**b**) Sintering curve of platinum paste.

**Figure 9 micromachines-13-00621-f009:**
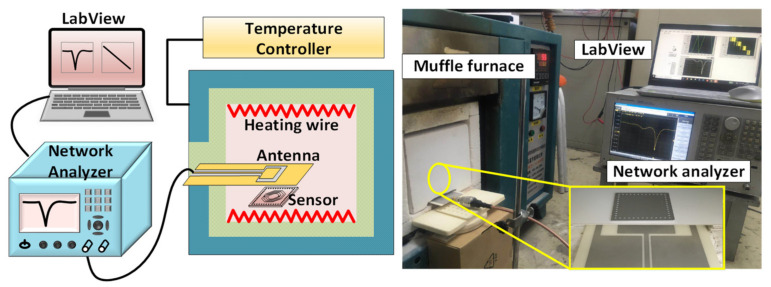
Schematic of the high-temperature testing platform.

**Figure 10 micromachines-13-00621-f010:**
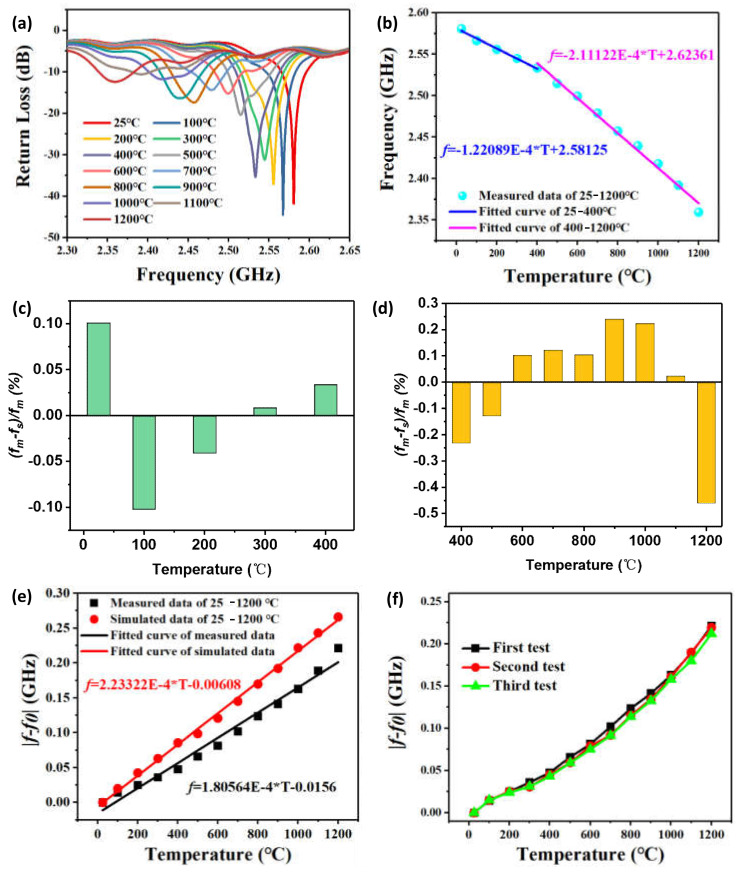
(**a**) The resonant frequency curves within the temperature of 25–1200 °C. (**b**) Linear fitting curves. (**c**) Nonlinearity errors in the temperature range of 25 to 400 °C. (**d**) Nonlinearity errors in the temperature range of 400 to 1200 °C. (**e**) Linear fitting curves of simulation and measurement. (**f**) Repeatable test.

**Table 1 micromachines-13-00621-t001:** The detailed parameters of the designed sensor.

Parameter	Value (mm)	Parameter	Value (mm)
a	1.5	L	28
b	1.4	W	28
t	7	D	1
g	0.3	p	2

## Data Availability

Not applicable.

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
