# Peer review of "Wireless Passive Microwave Antenna-Integrated Temperature Sensor Based on CSRR"

_micromachines, 2022, doi:10.3390/mi13040621_

Round 1
Reviewer 1 Report
The author has presented a complementary split ring resonator based in SIW. Whilst the text was well written, the following issues require the authors' attention:
1) I have no problem with the presentation. One of the issues I have is about the the novelty of this work. This kind of work has been done in many studies before. The following are just examples:
Jaehyurk Choi and Sungjoon,"Complementary Split Ring Resonator (CSRR)-Loaded Substrate Integrated Waveguide (SIW)", IEICE Trans. Commun., Vol., E95-B, No.1 January 2012.
S. Moitra et al., "Circular Complementary Split Ring Resonators (CSRR) based SIW BPF," 2019 Second International Conference on Advanced Computational and Communication Paradigms (ICACCP), 2019, pp. 1-5, doi: 10.1109/ICACCP.2019.8883003.
2) While the temperature monitoring can be considered as a good application, we still need a vector network analyzer to generate those scattering parameters. Unfortunately, almost none of the existing vector network analyzer is designed to work at such a high temperature. A vector network analyzer is also an expensive item. Perhaps, the author should propose a realistic solution to overcome this problem.
Reviewer 2 Report
The article is devoted to experimental studies of the new temperature sensor based on complementary split ring resonator (CSRR) is presented for ultra-high temperature applications. This is an interesting study. The article may be adopted with some changes:
1. In line 103 a formula has to be in the center.
2. The physical and geometric parameters of the sensor should be obtained so that the theoretical researchers have the ability to compare the numerical experiment results and the empirical experiment, made in this work.
Round 2
Reviewer 1 Report
The authors have given enough information on the nature of their work.